# Enhancing Robot Program Synthesis Through Environmental Context

**Tianyi Chen**
Fudan University
tianychen21@m.fudan.edu.cn

**Qidi Wang**
Fudan University
21210240038@m.fudan.edu.cn

**Zhen Dong**[†]
Fudan University
zhendong@fudan.edu.cn

**Liwei Shen**
Fudan University
shenliwei@fudan.edu.cn

**Xin Peng**
Fudan University
pengxin@fudan.edu.cn

## Abstract

Program synthesis aims to automatically generate an executable program that conforms to the given specification. Recent advancements have demonstrated that deep neural methodologies and large-scale pretrained language models are highly proficient in capturing program semantics. For robot programming, prior works have facilitated program synthesis by incorporating global environments. However, the assumption of acquiring a comprehensive understanding of the entire environment is often excessively challenging to achieve. In this work, we present a framework that learns to synthesize a program by rectifying potentially erroneous code segments, with the aid of partially observed environments. To tackle the issue of inadequate attention to partial observations, we propose to first learn an environment embedding space that can implicitly evaluate the impacts of each program token based on the precondition. Furthermore, by employing a graph structure, the model can aggregate both environmental and syntactic information flow and furnish smooth program rectification guidance. Extensive experimental evaluations and ablation studies on the partially observed VizDoom domain authenticate that our method offers superior generalization capability across various tasks and greater robustness when encountering noises.

## 1 Introduction

Program synthesis endeavors to produce an executable program that fulfills the prescribed specifications. Given that formulating a definitive and lucid specification can often prove more arduous than physically authoring the program, a widely-used method is to furnish *input/output* (I/O) examples as a close approximation of the requisite specification [16, 17, 27, 34], also known as Programming By Example (PBE). It allows a more straightforward presentation of the desired program functionality. The utilization of PBE enables machines to acquire the capacity to implicitly capture a set of regulations and generalize them to analogous scenarios. Over the past few years, PBE has demonstrated its remarkable potential to elevate automation and ease human labor in numerous domains, including but not limited to array transformation [3, 46, 56], graphic design [16], and comprehension of human demonstrations [21, 42].

In the realm of robot programs, a crucial factor is the interaction between the robot and its environment. In practice, a robot can solely rely on its sensors to perceive the environment by gathering diverse types of data, which forms the basis for every decision. Consider a robot designed for food delivery,

---

[†]Corresponding author

37th Conference on Neural Information Processing Systems (NeurIPS 2023).

which is tasked with transporting food to a specified destination. The robot is faced with a decision between two paths: one that is obstructed by barriers and leads to a dead-end, and another that is more circuitous and distant. If the robot is furnished with comprehensive knowledge of the environment, it may be possible to create a simple program consisting of only a few actions. However, if the robot can only obtain partial observations, such as RGB image data from its current perspective, a more intricate program with sophisticated control flow may be necessary to navigate out of the dead-end and reach the intended destination. Consequently, the latter program can generalize better in accomplishing more complex tasks. The variation in the level of observation can significantly impact the ultimate outcome. Therefore, the challenge in robot program synthesis arises from the fact that robots are limited to partial observations of their surroundings, making it difficult to assess the global impact of the generated program tokens toward the desired output. This is due to the fact that a comprehensive understanding of the environment is not achievable.

At present, there exists a diverse range of deep learning techniques aimed at solving PBE problems using neural network models [5, 8, 32]. Empirical studies [3, 13] have shown that recurrent neural network (RNN) architectures can learn strategies that generalize across problems and remain robust to moderate levels of noise. However, these methods have limitations in their ability to effectively utilize environmental observations due to their reliance on the RNN as the backbone. Additionally, recent studies have introduced reinforcement learning [7, 28], debugger method [19] and latent generative methods [10, 48] into the field. Furthermore, other studies have revealed that incorporating intermediate states produced by executing partially generated programs can improve the capability to address program synthesis tasks, especially for sequential programming tasks [9, 43, 47, 56]. These techniques have primarily been evaluated on the Karel [12, 18, 44] dataset, in which the model possesses complete awareness of the entire environment. To clarify, even if a particular location in the environment is inaccessible or unknown to the robot, the model can still generate a program with knowledge of that spot. Nonetheless, this assumption of acquiring a global view of the environment is arduous to achieve in reality, and their aptitude to generate accurate programs with only partial observations remains to be explored.

To tackle this issue, we introduce our novel **E**nvironmental-context **V**alidated l**A**tent **P**rogram **S**ynthesis framework (EVAPS). Drawing inspiration from the successful previous work SED [19], as well as the current trend of large language models [1, 22, 23], we believe that the trail-eval-repair loop provides valuable guidance for the evolution of programs, leading to better generalization ability. In order to leverage partial environmental observations, EVAPS initially obtains candidate programs by employing the same neural program synthesizer component used in SED. Subsequently, by executing the candidate program, EVAPS acquires the partial environment before and after each action (*i.e.,* the environmental context). By concurrently modeling both the environmental and syntax contexts, EVAPS iteratively alters the programs that do not produce the correct output in an effort to resolve semantic conflicts across program tokens, ultimately correcting erroneous program fragments.

The framework was assessed in the partially observed VizDoom domain [24, 55], where a robot operates in a 3D world and interacts with its environment, including objects and adversaries. The results of the experiment demonstrate that the proposed approach is superior in modeling program semantic subtleties and resolving potential errors, thereby enhancing its generalization ability. Additionally, we authenticate the efficacy of leveraging environmental contexts and aligning them with code syntax by carrying out an ablation study. Further experiments show that our method is more capable of solving complex tasks compared to prior works. Furthermore, we validate that our method remains robust when encountering various levels of noise.

## 2    Problem Formulation

In a typical PBE task, we are given a set of $N$ training samples, represented as $\left\{ \{IO\}^K, \mathcal{P} \right\}^N$. Each sample consists of $K$ I/O pairs that act as specifications, and a gold program $\mathcal{P}$ that correctly maps the input state $I$ to the output state $O$, such that $\mathcal{P}(I_i^k) \rightarrow O_i^k$, where $\forall i \in [1, N], \forall k \in [1, K]$. Previous studies have investigated techniques to synthesize a candidate program $\hat{\mathcal{P}}$ solely based on I/O. Our objective is to train a neural model, denoted as $\mathcal{G}$, which takes a candidate program $\hat{\mathcal{P}}$, and the corresponding environmental contexts $\mathcal{O}$ obtained by executing the program $\hat{\mathcal{P}}$ as an input. The model generates a refined program by incorporating environmental contexts: $\mathcal{G}(\hat{\mathcal{P}}, \mathcal{O}) \rightarrow \mathcal{P}^*$, such that the mapping of program $\mathcal{P}^*(I)$ approaches more closely to $\mathcal{P}(I)$ given input $I$. To assess the

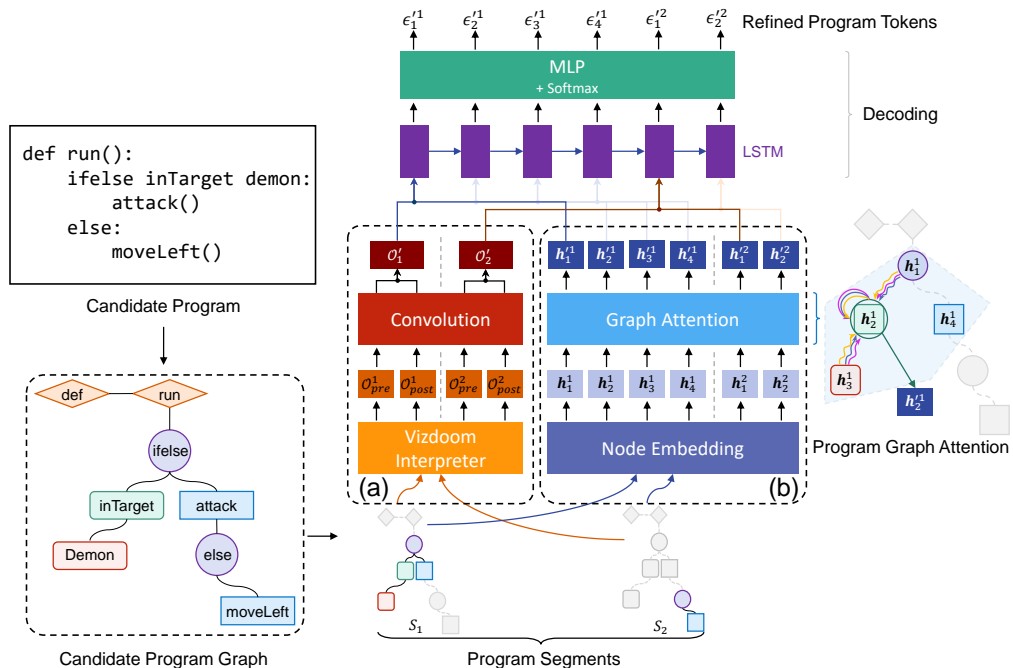

Figure 1: The EVAPS architecture proposed involves the preprocessing of the candidate program into a program graph, which can be further utilized for code segment generation and code symbol representation learning. The encoder comprises (a) the *partial observation leveraging* module that utilizes insights gleaned from partial observations by learning latent environmental context representations, and (b) the *code symbol alignment* module that brings richer semantic features by aggregating neighbor syntax symbol representations. Eventually, the decoder outputs refined program tokens by incorporating both modules.

generalization ability of the model $\mathcal{G}$, we include another set of held-out examples that are not part of the training samples, denoted as $\{IO\}^{K_{test}}$, enabling us to measure the equivalence between $\mathcal{P}^*$ and $\mathcal{P}$ using $IO^k \in \{IO\}^K \cup \{IO\}^{K_{test}}$.

## 3 Methodology

### 3.1 Leveraging Partial Observations

In the VizDoom domain, the robot perceives its surroundings $\mathcal{O}$ by collecting an RGB image $\mathcal{O}_R \in \mathbb{R}^{n_H \times n_W \times n_C}$ and a depth buffer $\mathcal{O}_D \in \mathbb{R}^{n_H \times n_W \times 1}$ from its visual perspective, denoted as $\mathcal{O} = \langle \mathcal{O}_R, \mathcal{O}_D \rangle$. This forms its partial observation of the world. By executing a program, we can obtain a sequence of observation changes $\mathcal{O}_0 \overset{ac_1}{\Longrightarrow} \mathcal{O}_1 \overset{ac_2}{\Longrightarrow} ... \overset{ac_{n_e}}{\Longrightarrow} \mathcal{O}_{n_e}$, where $ac_i$ refers to the $i$-th executed action, $n_e$ denotes the number of total execution steps, and $\mathcal{O}_0$ denotes the initial observed state. As depicted in Figure 1(a), the hidden environment representations can be learned by applying convolutional network layers, and the embedded vector at the $l$-th layer can be denoted as:

$$\mathcal{O}_{x,y}^{[l]} = \text{Conv}(\mathcal{O}_{x,y}^{[l-1]}, \mathbf{K}) = g^{[l]}\left(\sum_{h=1}^{n_H^{[l]}} \sum_{w=1}^{n_W^{[l]}} \sum_{c=1}^{n_C^{[l]}} \mathbf{K}_{h,w,c} \cdot \mathcal{O}_{x+h-1,y+w-1,c}^{[l-1]}\right), \tag{1}$$

where $\mathbf{K}$ represents the convolutional kernel, and $g^{[l]}$ signifies the activation function at layer $l$.

To effectively utilize insights gleaned from partial observations, it is essential that we obtain the antecedent states $\mathcal{O}_{pre}$ (also known as the precondition) and the potential consequent effects $\mathcal{O}_{post}$ corresponding to each token of program $\hat{\mathcal{P}}$, so that the model can implicitly evaluate whether this token contributes towards completing the task in the current context, and determine whether substitution with another token would heighten the likelihood of task fulfillment.

Given that only those tokens that denote robot actions (*e.g.,* `moveForward`) have the potential to alter the environment, whereas other tokens function as constituents of a particular statement (*e.g.,* `if` and `while`), a straightforward token-by-token modification strategy appears unappealing. Since non-action tokens cannot exist independently and can only render a code block semantically complete when they co-occur with action tokens, we deem it more fitting to process them as a unified segment.

We define a segment as a triplet comprising of $S = \langle cond, token, body \rangle$. Here, $cond$ refers to the condition of the segment, and $body$ denotes the associated actions or subsegments. In the case of an action token, the segment form deteriorates into $S = \langle null, action, null \rangle$. By recursively parsing the abstract syntax tree of program $\hat{\mathcal{P}}$, we can eventually yield a collection consisting of $T$ program segments, denoted as $S_T$. And by executing program segment $S_t \in S_T$ using the VizDoom interpreter, we can acquire the post-segment effect with $Execute(S_t, \mathcal{O}_{pre}) \to \mathcal{O}_{post}$, thereby obtaining the environmental context $\langle \mathcal{O}_{pre}^t, \mathcal{O}_{post}^t \rangle$ of segment $S_t$, presented in Figure 1. The interconnection between the observable world state change and the code segment promotes greater semantic versatility, which, in turn, enables the model to calculate the probability of substituting each token based on the partial environment, resulting in the following formulation:

$$p_\theta(\mathcal{P}_m^* | \mathcal{O}_{pre}, \mathcal{O}_{post}) = \prod_{S_t \in S_T} \left( \prod_{s=1}^{L_t} p_\theta \left( \epsilon_s' | \epsilon_1, ..., \epsilon_{s-1}, \epsilon_{s+1}, ..., \epsilon_{L_t}, \mathcal{O}_{pre}^t, \mathcal{O}_{post}^t \right) \right), \quad (2)$$

where $L_t$ denotes the number of tokens in segment $S_t$, $\epsilon_s$ refers to the $s$-th token of segment $S_t$, and $\epsilon_s'$ represents the refined program token that substitutes $\epsilon_s$. To optimize the model parameters, the objective is to minimize the cross-entropy loss, given $N$ training examples, where $S_T^i$ represents the program segment set of the $i$-th example, which contains $T_i$ segments:

$$\mathcal{L} = -\frac{1}{N} \sum_{i=1}^{N} \frac{1}{T_i} \sum_{S_t \in S_T^i} \frac{1}{L_t} \sum_{s=1}^{L_t} \log \left[ p_\theta(\epsilon_s' | \epsilon_1, ..., \epsilon_{s-1}, \epsilon_{s+1}, ..., \epsilon_{L_t}, \mathcal{O}_{pre}^t, \mathcal{O}_{post}^t) \right]. \quad (3)$$

## 3.2 Code Symbol Alignment

Now that the environmental contexts have been obtained, they are currently paired separately with their own code segments, and the correlations among environmental contexts are not fully exploited. Relying solely on partial observation is insufficient for the model to comprehend the semantic and syntactic long-range connections of program tokens, making it arduous for the model to consolidate information that traverses through subsequent segments during the program repair process. Furthermore, with regard to program syntax, the inherent nature of a program necessitates the model to jointly reason over code symbols, such as types. For instance, the code `while(turnLeft)` is syntactically incorrect because `turnLeft` is an action rather than a perception, which is prohibited to follow `while`. Therefore, it is desirable to align the code symbols with the partial observations to facilitate a more comprehensive information flow.

To accomplish this, we first construct a program graph $G \subseteq V \times E$ (presented in Figure 1) to represent each candidate program $\hat{\mathcal{P}}$, where $V$ denotes the set of program token nodes and $E$ denotes the set of undirected edges. Subsequently, we employ graph attention [6, 50] for each token to capture a contextualized representation. Similar to the definition of a segment, each node within the program graph possesses a leftward pointer that directs towards nodes signifying conditions, and a rightward pointer that indicates a subsequent token node. The node's value encompasses the token index, and token type. Meanwhile, the relationship between tokens and their corresponding segments is stored. In this way, we not only align the code symbols and environmental contexts but also aggregate further semantic and syntactical representations. Additionally, it enables the model to effectively capture the nested structure of a program, thereby bringing more expressiveness.

To elaborate, we begin by obtaining the node representations $\boldsymbol{h}_i \in \mathbb{R}^d$ for each node $V_i \in G$ through embedding. Subsequently, a scoring function $\alpha : \mathbb{R}^d \times \mathbb{R}^d \to \mathbb{R}$ is employed to compute the attention score of its neighboring node $V_j$. This score reflects the significance of a neighbor node. The attention scores are then normalized across all neighbors $j \in \mathcal{N}_i$ using softmax, which is defined as:

$$\alpha(\boldsymbol{h}_i, \boldsymbol{h}_j) = \frac{\exp\left(\boldsymbol{A}^\top \text{LeakyReLU}(\boldsymbol{W} \cdot [\boldsymbol{h}_i \| \boldsymbol{h}_j])\right)}{\sum_{k \in \mathcal{N}_i \cup \{i\}} \exp\left(\boldsymbol{A}^\top \text{LeakyReLU}(\boldsymbol{W} \cdot [\boldsymbol{h}_i \| \boldsymbol{h}_k])\right)}, \quad (4)$$

where $\boldsymbol{A}$ and $\boldsymbol{W}$ are weight matrices that can be learned, and $\|$ represents vector concatenation. As a result, node $V_i$ acquires a new representation by calculating the weighted mean of the modified features of its neighboring nodes, as illustrated in Figure 1(b):

$$\boldsymbol{h}_i^{'} = \alpha_{i,i} \cdot \boldsymbol{W}\boldsymbol{h}_i + \sum_{j \in \mathcal{N}_i} \alpha_{i,j} \cdot \boldsymbol{W}\boldsymbol{h}_j. \tag{5}$$

Next, we can obtain the corresponding environmental embedding of the token in $V_i$ through $\mathcal{O}_i^{'} = \mathrm{Conv}(\mathcal{O}_i, \mathbf{K})$ (Eq. (1)). To align the code symbol representations and environmental contexts, we update the representation of node $V_i$ by simply concatenating these embeddings $\mathbf{x}_i = [\boldsymbol{h}_i^{'}\|\mathcal{O}_i^{'}]$. This leads to a modification to Eq. (2):

$$p_\theta(\mathcal{P}_m^*|\mathcal{O}_{pre}, \mathcal{O}_{post}) = \prod_{S_t \in S_T} \left( \prod_{s=1}^{L_t} p_\theta\left(\epsilon_s^{'}|\mathbf{x}_s\right) \right). \tag{6}$$

The key connection between the proposed approach and observable environments resides in establishing a connection between the program execution context and the partially observable environment. EVAPS takes the execution context of segment $S$ and the corresponding partially observable environment as a unit for model training. This enables the combination of program syntax and the corresponding partially observable environment to predict a token, thereby enhancing the accuracy of program synthesis. As a result, the model can adeptly resolve the underlying conflict within a program by concurrently capturing well-integrated and highly pertinent semantic and contextual patterns, culminating in a greater practicality of programs.

## 4 Experiment

### 4.1 Experimental Setup

**VizDoom Domain.** The programs in this study are structured using Domain-Specific Languages (DSL) in the VizDoom environment [24]. This environment provides a 3D world that closely resembles real-world scenarios, allowing the robot to efficiently perceive, interpret, and learn the 3D realm. In recent times, there has been a burgeoning interest in VizDoom exploring the robot's capabilities to make tactical and strategic determinations [2, 11, 14, 25, 41, 47, 53]. The VizDoom experimental environment consists of 7 action primitives, 6 perception primitives, and the state representation dimension is $120 \times 160 \times 3$. The average number of steps required to complete tasks is $4.6$. We contend that the VizDoom environment is more suitable for robot program synthesis research compared to Karel, due to its limited partially observed nature [55], which renders the decision and execution process more realistic and practical for the robot. The DSL comprises actions, perceptions, and control flows, sufficient to encapsulate the fundamental concepts of robot programming. The detailed description is presented in Figure 2.

Program $m ::= $ def run() : $s$
Statement $s ::= $ while($b$) : ($s$) | $s_1$; $s_2$ | $a$ | repeat($r$) : ($s$) |
    | if($b$) : ($s$) | ifelse($b$) : ($s_1$) else : ($s_2$)
Repetition $r ::= $ Integer number of repetitions
Condition $b ::= $ perception $\gamma$ | not $b$
Action $a ::= $ moveForward | moveBackward | moveLeft
    | moveRight | turnLeft | turnRight | attack
Perception $\gamma ::= $ isThere $\varepsilon$ | inTarget $\varepsilon$
Monster $\varepsilon ::= $ demon | hellKnight | revenant

Figure 2: The Domain-Specific Language (DSL) for Vizdoom programs comprises action primitives, perception primitives and control flows.

**Dataset.** To generate a dataset for learning environmental-context embeddings, we adopt the same approach as previous studies [14, 47], randomly generating $100,000$ distinct samples. The dataset is then partitioned into a training set with $70,000$ samples, a validation set with $20,000$ samples, and a testing set with $10,000$ samples. We adhere to the program synthesis conventions [7, 19], where each sample includes a correct program, $5$ input-output pairs that act as specifications, and an extra input-output example as the test sample, which is solely employed for evaluation purposes and not for training. During the generation process, we also conduct checks to ensure that all execution branches in the program are covered at least once, thereby representing all aspects of the program's behavior and providing sufficiently challenging tasks.

Table 1: The average accuracy (with standard deviation) of all methods evaluated on three metrics across Vizdoom tasks, assessed over 5 random seeds. The **best results** are highlighted in bold.

| | Methods | Exact Match | Semantic Match | Generalization |
|---|---|---|---|---|
| Top-1 | LGRL [7] | 2.18% (0.34%) | 33.42% (1.12%) | 31.49% (1.26%) |
| | SED [19] | 12.69% (0.55%) | 43.07% (0.57%) | 38.63% (0.51%) |
| | Inferred Trace [43] | 14.73% (1.73%) | 36.87% (1.81%) | 35.09% (1.73%) |
| | Latent Execution [10] | 3.53% (1.12%) | 44.40% (2.31%) | 42.25% (2.22%) |
| | Transformer [49] | 15.53% (1.75%) | 26.51% (1.96%) | 25.93% (1.77%) |
| | **EVAPS (Ours)** | **36.40%** (2.52%) | **50.29%** (2.18%) | **48.90%** (2.23%) |
| Top-5 | LGRL [7] | 4.58% (0.43%) | 54.25% (1.01%) | 51.56% (1.08%) |
| | SED [19] | 25.48% (0.89%) | 66.76% (0.92%) | 62.25% (0.65%) |
| | Inferred Trace [43] | 27.49% (1.87%) | 55.13% (0.59%) | 52.80% (0.88%) |
| | Latent Execution [10] | 4.91% (1.32%) | 50.76% (2.07%) | 48.62% (1.81%) |
| | Transformer [49] | 34.80% (4.23%) | 47.96% (2.24%) | 46.76% (2.13%) |
| | **EVAPS (Ours)** | **50.62%** (2.07%) | **69.82%** (1.72%) | **67.82%** (1.81%) |
| Top-20 | LGRL [7] | 9.16% (0.76%) | 66.22% (0.19%) | 63.49% (1.06%) |
| | SED [19] | 51.23% (1.01%) | 71.42% (3.66%) | 67.06% (3.76%) |
| | Inferred Trace [43] | 37.75% (2.10%) | 66.69% (1.82%) | 63.96% (1.54%) |
| | Latent Execution [10] | 9.16% (1.21%) | 59.93% (1.96%) | 57.53% (1.50%) |
| | Transformer [49] | 51.53% (3.61%) | 64.69% (2.90%) | 63.09% (2.77%) |
| | **EVAPS (Ours)** | **62.22%** (3.28%) | **79.20%** (1.08%) | **76.84%** (1.13%) |

**Evaluation Metric.** To comprehensively assess the capability of synthesizing programs based on provided inputs and outputs, we evaluate all models using three metrics: (1) *Exact Match*. It considers a generated program to be an exact match if every token in it is identical to the gold program. This is expressed mathematically as: $\sum_{i=1}^{N} \Phi[\mathbf{TKN}_t(\mathcal{P}_i^*) = \mathbf{TKN}_t(\mathcal{P}_i)], \forall t \in [1, L]$ where $\mathbf{TKN}_t(\mathcal{P}_i)$ refers to the $t$-th token of program $\mathcal{P}_i$, $\Phi[\cdot]$ yields 1 if the condition is satisfied and 0 otherwise, and $L$ denotes the length of the program. (2) *Semantic Match*. It takes into account that a synthesized program may be semantically equivalent but not syntactically identical to the gold program. This metric is expressed as: $\sum_{i=1}^{N} \Phi[\mathcal{P}_i^*(I_i^k) = \mathcal{P}_i(I_i^k) = O_i^k], \forall I_i^k, O_i^k \in \{IO\}_i^K$. (3) *Generalization Match*. To genuinely evaluate the generalization ability of a model, it is crucial to assess its ability to generalize on unseen samples in addition to semantic consistency. To achieve this, we include additional examples and evaluate the model's performance as: $\sum_{i=1}^{N} \Phi[\mathcal{P}_i^*(I_i^k) = \mathcal{P}_i(I_i^k) = O_i^k], \forall I_i^k, O_i^k \in \{IO\}_i^K \cup \{IO\}_i^{K_{test}}$. Kindly note that our primary concern regarding the model's performance is the metric of generalization match.

## 4.2 Results

**Overall Synthesis Performance.** Table 1 presents the primary outcomes of our EVAPS in comparison to prior works [7, 10, 43, 49] for VizDoom program synthesis. Notably, we have implemented the beam search technique for Inferred Trace, Latent Execution, and Transformer to enable the generation of multiple candidate programs while preserving their original architecture. Additionally, we have incorporated a convolutional neural network on top of the Transformer to facilitate the learning of specification embedding, given that the specification is in the form of grids rather than sequences, and the Transformer serves as the decoder. The supplementary material contains the detailed parameters of all the compared methods.

We can observe that EVAPS consistently outperforms other methods across three metrics, with an approximate improvement of $+9.6\%$ compared to the runner-up. This indicates its superior capability to resolve potential semantic errors by incorporating environmental contexts, thereby enhancing its generalization ability while preserving syntax modeling ability. Moreover, EVAPS surpasses SED primarily due to the fact that SED only enhances program tokens based on execution outcomes and

Table 2: The average accuracy (with standard deviation) of all ablation methods evaluated on three metrics across Vizdoom tasks, assessed over 5 random seeds. The **best results** are highlighted in bold, and Gen.$\Delta$ represents the mean improvement in generalization match.

|  | Methods | Exact Match | Semantic Match | Genralization | Gen.$\Delta$ % |
|---|---|---|---|---|---|
| Top-1 | Naïve | 3.42% (1.80%) | 32.29% (5.41%) | 30.29% (5.36%) | - |
|  | EVAPS+O | 23.56% (5.02%) | 46.80% (2.53%) | 45.24% (2.49%) | +14.95% |
|  | EVAPS+S | 23.42% (1.84%) | 42.47% (2.87%) | 41.09% (2.79%) | +10.80% |
|  | **EVAPS** | **36.40%** (2.52%) | **50.29%** (2.18%) | **48.90%** (2.23%) | +18.61% |
| Top-5 | Naïve | 7.38% (3.15%) | 52.11% (1.10%) | 49.60% (1.50%) | - |
|  | EVAPS+O | 40.15% (4.52%) | 61.49% (1.85%) | 59.71% (1.72%) | +10.11% |
|  | EVAPS+S | 40.90% (4.80%) | 58.18% (3.42%) | 56.58% (3.20%) | +6.98% |
|  | **EVAPS** | **50.62%** (2.07%) | **69.82%** (1.72%) | **67.82%** (1.81%) | +18.22% |
| Top-20 | Naïve | 12.87% (3.25%) | 64.80% (1.56%) | 61.96% (1.57%) | - |
|  | EVAPS+O | 55.35% (4.90%) | 71.60% (2.54%) | 69.63% (2.22%) | +7.67% |
|  | EVAPS+S | 56.29% (5.26%) | 69.85% (2.79%) | 67.78% (2.56%) | +5.82% |
|  | **EVAPS** | **62.22%** (3.28%) | **79.20%** (1.08%) | **76.84%** (1.13%) | +14.88% |

specifications, whereas EVAPS places greater emphasis on the semantic locality, supplemented by utilizing partial observations. While the attention mechanism of Transformer facilitates its ability to learn syntax, its generalization ability is relatively poor. Although LGRL can achieve comparable results in terms of generalization, it faces challenges in producing program sequences that are identical to the gold programs due to its tendency to generate semantically equivalent programs in a more complex environment. Inferring traces can provide information gain, but the partially observed environment restricts the model from capturing subtle semantic features. This limitation also applies to the latent executor, as the global status becomes agnostic to the model, thereby limiting its ability to extrapolate the execution states and learn the latent effects of each token.

**Ablation Study.** To thoroughly verify the efficacy of the *partial observation leveraging* module and the *code symbol alignment* module of EVAPS, we perform an ablation study to authenticate their capabilities and scrutinize their impacts as Table 2 presents. We consider the following ablations:

- Naïve: a variation of the program synthesis baseline LGRL [7] that utilizes greedy decoding (*i.e.,* implements a beam search with beam size $B = 1$). It employs a standard encoder-decoder architecture that acquires the ability to directly synthesize a program from scratch.
- EVAPS+O: an ablation of EVAPS incorporating partial environmental observations to modify tokens and resolve semantic conflicts in synthesized programs.
- EVAPS+S: an ablation of EVAPS aggregating long-range semantic and syntax connections of neighboring code symbols.
- EVAPS (EVAPS+O+S): incorporates both modules, allowing for a more comprehensive information flow and leveraging environmental context to facilitate program synthesis.

Based on the ablation outcomes presented in Table 2, we can deduce that utilizing only partial environmental observations or code symbol alignment can enhance the joint semantic and syntax modeling ability. However, the information gain is limited if only one of these techniques is employed. This is because the former concentrates primarily on the environmental context of a particular token, disregarding the implicit connections between different code segments in semantic spaces. On the other hand, the latter focuses mainly on information aggregation rather than environmental contexts. Consequently, combining these techniques can endow the model with supplementary information and further boost the program synthesis process, resulting in an approximate enhancement of $+16\%$ in terms of generalization.

**Task Complexity.** Considering the varying complexities of different robot tasks, generating corresponding codes to accomplish them may pose varying levels of difficulty. Hence, we endeavor to delve into the efficacy of each method in handling diverse levels of complexity. The issue of program aliasing, wherein two programs are semantically equivalent but not identical (e.g., `while(r=3):turnLeft` and `turnRight`), renders the evaluation of complexity using program

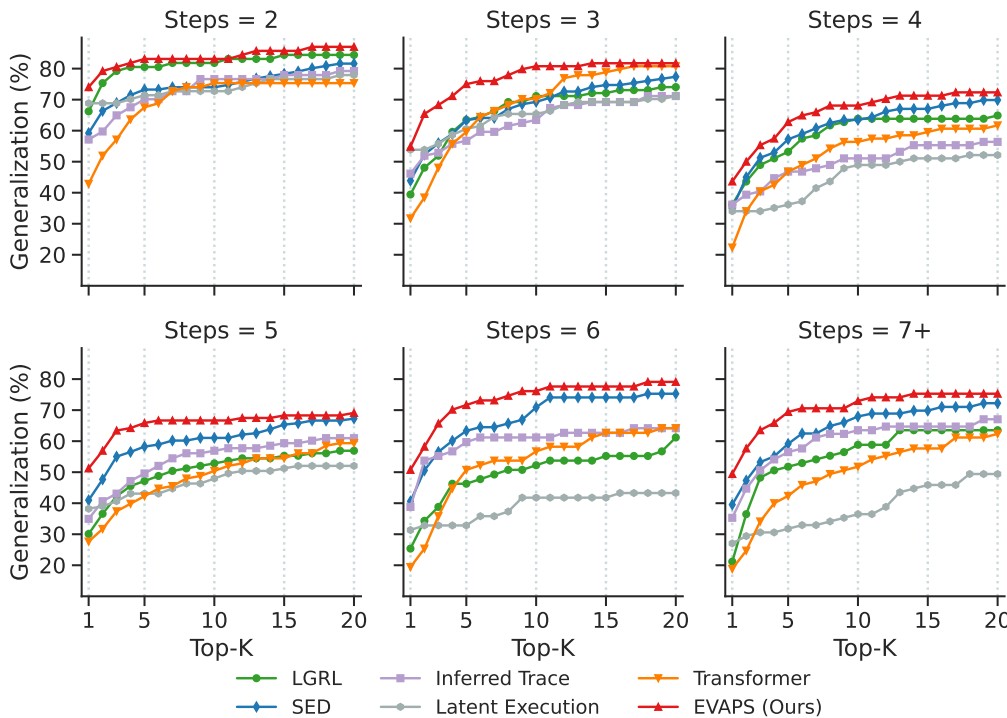

Figure 3: The generalization capability of all methods ranging from Top-1 to Top-20 evaluated across six categories that were partitioned based on the minimum number of steps required to complete the task. The tasks that can be accomplished with at least seven steps were denoted as 7+.

length or control flow presence imperfect. Hence, we propose partitioning the tasks into different complexity levels based on the *minimum number of steps required to complete the task*, which we denote as steps.

To be specific, we have divided the testing set into six distinct categories based on their level of complexity, with each category requiring 2, 3, 4, 5, 6, or 7+ steps to complete, respectively. The 7+ category is reserved for tasks that demand no fewer than seven steps to accomplish. We then proceed to train and evaluate the model's performance in each category separately, and the results of this analysis are presented in Figure 3. Firstly, it is evident that as the complexity of the task increases, all methods experience a certain degree of performance degradation. This is reasonable since the embedding space becomes more intricate to learn as the task becomes more complex. Furthermore, we can infer that the performance gap between EVAPS and other methods widens as the complexity increases. This suggests that EVAPS is better equipped to adapt to more intricate tasks, owing to its ability to break down complex objectives into simpler behaviors by leveraging partial environmental contexts. On the other hand, the modeling ability of Latent Execution becomes relatively inferior, particularly for Step=7+, which indirectly implies that it is challenging for a model to deduce the impact of complex behaviors in a single attempt.

Interestingly, for most techniques, the ability to generalize experiences a surge at approximately the Top-5 candidate programs. However, the rate of improvement tends to decelerate and the curve becomes flat. We surmise that this phenomenon occurs because certain semantic representations are conspicuous and uncomplicated to acquire, whereas others are more nuanced. Even with more candidate programs, the semantic features corresponding to these programs are somewhat overlapped, thereby reaching the limit for capturing semantic subtleties and hindering further generalization.

**Handling Noisy Observations.** For robot programming, real-world observations are collected through sensors in real-time, making it inevitable for occasional noise to occur. Consequently, we tend to evaluate the ability of our method to handle such noise. Given that the VizDoom dataset lacks noisy examples, we artificially introduce noise into the dataset by invalidating a certain percentage of observation grids based on uniform random probability [15].

The results regarding different levels of noise (with a maximum noise of 20%) are illustrated in Figure 4. It can be inferred that the presence of noisy observations inevitably undermines the capacity of EVAPS to learn program semantics and generalize. Nevertheless, EVAPS still manages to exhibit a decent modeling ability. If we consider all Top-K potential candidates, the maximum performance drop obtained is merely about 7.5%. Meanwhile, with fewer candidate programs, the performance gap widens, but it is bound to be 12%. This outcome aligns with our expectations, thus we can conclude that EVAPS is capable of recognizing significant semantic patterns and maintaining robustness when encountering noises.

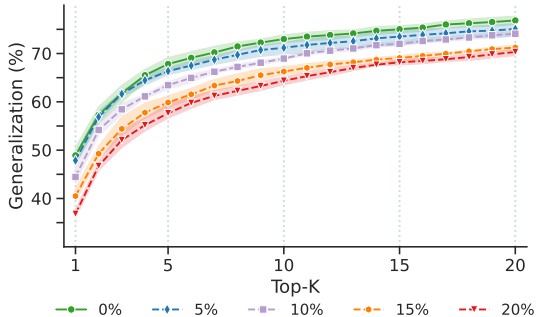

Figure 4: The generalization capacity of EVAPS in the presence of diverse levels of noise, accompanied by a 95% confidence interval band.

**Dicussion.** The partial observation that the proposed approach relies on is often available in practice, thus it is theoretically feasible to apply this approach to the real world. Certainly, we anticipate some practical difficulties during the transition from the simulation environment to the real world. For instance, it can be labor-intensive to collect enough environmental data from the real world for the purpose of training. Further investigation into the efficient acquisition of environmental data while sustaining its quality is yet to be explored in future works.

## 5  Related Work

**Programming By Example.** In the realm of program synthesis, the synthesis of programs from examples has long been a preoccupation of researchers. In recent years, deep learning methods have made many breakthroughs [37, 13, 36], and many researchers have proposed different methods for Improve the accuracy of synthesis. Symbolic search is regarded as an efficient technique for addressing program synthesis challenges [40, 38, 51, 26]. Wang et al. have devised a language for abstract queries, which facilitates the generation of SQL [51]. Moreover, Spoc has recognized the significance of pseudocode and explores its potential for aiding code search, focusing on alternative translations of pseudocode [26]. While symbolic search can yield promising outcomes within specific training domains, it lacks the ability to expand, and it is relatively hard to generate code snippets beyond the training datasets.

To tackle the issue of code fragment homogeneity, two approaches have gained prominence: the utilization of hidden variables and the employment of reward mechanisms. In order to effectively utilize the information contained in latent variables, some researchers consider using latent generative models [3, 10, 48]. Aim at obtaining the intermediate execution results of partial programs in C language, Chen et al. approximate by learning the latent representation of partially generated programs to facilitate program token search [10]. Additionally, reinforcement learning has been integrated into synthetic neural network training in numerous studies [7, 16, 9, 48, 19]. Bunel et al. combined program synthesis with reinforcement learning [7], and avoided the problem of Program Aliasing by defining proper rewards; Trivedi et al. considered a two-stage learning scheme on this basis [48]. First, a program embedding space is learned in an unsupervised manner, and then the most suitable program is searched in the embedding space with a given reward.

In our approach, we also use a reward mechanism for generating candidate programs. However, we extend this by adopting the concept from SED [19] to train an evaluator that assesses the candidate programs for further repair. While both EVAPS and SED incorporate environmental observation to train the neural program debugger, there are notable distinctions between the two: i) The execution feedback of SED relies on a global perspective, which is only available in some special cases. Conversely, EVAPS embraces partial observation, which is more achievable in real-world scenarios. ii) SED treats the program as a whole for training, while EVAPS pays more attention to the execution context. iii) EVAPS establishes a connection between the program execution context and the partial observable environment. Distinguishing itself from SED, EVAPS places greater emphasis on the partially observable environment in which the robot operates and the local context of the program.

We believe that this approach enables the network to concentrate more precisely on the actual scope of the impact of the program.

**Program Repair.** Program repair has also been a widely researched topic. Most researchers pay attention to the grammatical features [30, 29, 45] and semantic information [52, 54] of programs. Rolim and Hua [39, 20] employ abstract syntax trees to derive effective outputs, while Tfix [4] treats programs as text and trains a general Transformer model for repairs. While these approaches have achieved breakthroughs in different problem domains, they often overlook the semantic aspects of programs. Wang et al. address this by incorporating the learning of semantic embeddings from program execution trajectories [52], while Ye et al. enhance neural networks using a loss function based on program compilation and execution information [54]. Our work takes both semantic and grammatical information into account. Our evaluation network, which is based on the execution information of the program in the environment and the program's grammar domain-specific language, can effectively accomplish program repairs.

Regarding traditional program repair approaches, we are aware of a branch of rich works in the program repair area, which can be classified into three categories: i) Search-based. This type of approach considers program repair as a search problem, exploring the space of all possible program candidates to identify one that satisfies the given weak specification, i.e., test cases. One of the prior works in this area is GenProg [31]. ii) Semantic-based. This type of approach extracts semantic information from the program under repair (typically represented as path constraints) and then generates patches by solving those constraints. A well-known work is SemFix [35]. iii) Learning-based. This type of approach leverages a number of patches generated by developers to learn a model that repairs programs. An earlier work is Prophet [33]. Those approaches depend on high-quality test suites to validate patch candidates, which are unavailable in the setting our approach is targeted at, i.e., VizDoom.

## 6   Conclusion

This paper introduces EVAPS, a novel approach to augment the applicability of neural program synthesis by integrating partial environmental observations. By utilizing both the *environmental context leveraging* module and the *code symbol alignment* module, the ability to rectify semantically erroneous program segments and generalize across various tasks is substantially enhanced. Furthermore, we verify that EVAPS exhibits greater resilience when confronted with noise and can adeptly capture subtle features even under more intricate tasks. The proposed approach presents a promising direction for robot program synthesis by incorporating partial environmental observations. The ability of the framework to rectify semantically erroneous program segments, generalize across tasks, and maintain robustness in the presence of noise showcases its potential for practical applications in robot program synthesis.

## Acknowledgments and Disclosure of Funding

This work was supported by Shanghai Municipal Science and Technology Major Project (No.2021SHZDZX0103).

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
