# Supplementary Material for
# Enhancing Robotic Program Synthesis Through Environmental Context

## A    Implementation Details

### A.1    Hardware and Software Configurations

All experiments were conducted on Ubuntu 20.04.5 LTS (Linux version 5.15.0-46-generic) utilizing Python 3.9.0, PyTorch 1.12.1 [9], and PyTorch-Geometric 2.3.0 [5]. The hardware employed consisted of 24 Intel(R) Xeon(R) Gold 5317 CPUs @ 3.00GHz, 8 modules of 32GB memory (with a speed of 3200MT/s), and 2 NVIDIA A40 GPUs with 48GB of memory each (NVIDIA UNIX x86_64 Kernel Module 510.108.03, CUDA version 11.6, cuDNN version 8.3).

### A.2    Network Architecture

For the program synthesizing stage, the structure of the I/O encoder is elaborated in Table 1, where we employ $d_{k_1} \times d_{k_2}$-$s$-$d_o$ Conv to denote the 2D convolution with kernel size $d_{k_1} \times d_{k_2}$, stride $s$, and output channel $d_o$. Additionally, BN refers to batch normalization [8], and $d_i$-$d_o$ Linear denotes the fully-connected layer with input feature $d_i$ and output feature $d_o$. The I/O encoder utilizes residual networks [7] and takes I/O pair with size $5 \times 5 \times 3$ as inputs.

Table 1: The structure of the I/O encoder for synthesizing stage.

| Layers | Output |
|---|---|
| $3 \times 3$-1-32 Conv BN LeakyReLU | $5 \times 3 \times 32$ |
| $3 \times 3$-1-32 Conv BN LeakyReLU | $5 \times 3 \times 32$ |
| $3 \times 3$-1-64 Conv | $5 \times 3 \times 64$ |
| $3 \times 3$-1-64 Conv | $5 \times 3 \times 64$ |
| $3 \times 3$-1-64 Conv BN LeakyReLU | $5 \times 3 \times 64$ |
| $3 \times 3$-1-64 Conv | $5 \times 3 \times 64$ |
| $3 \times 3$-1-64 Conv | $5 \times 3 \times 64$ |
| $3 \times 3$-1-64 Conv BN LeakyReLU | $5 \times 3 \times 64$ |
| 960-512 Linear | 512 |

To improve candidate programs through environmental contexts, the decoder's structure is elaborated in Table 2. Here, we utilize $d_o$-$h$ GATv2Conv to represent the dynamic graph attention variant [1] with output channel $d_o$ and multiple attention heads $h$, and $d_o$-$n_l$ denotes the $n_l$ layered bi-directional LSTM with output feature $d_o$. Additionally, $|\mathcal{V}|$ refers to the size of the Vizdoom DSL vocabulary and $L_t$ denotes the length of a candidate program. The decoder receives the environmental context, which comprises a depth buffer with dimensions of $30 \times 40 \times 15$ and an RGB automap buffer

with dimensions of $30 \times 120 \times 15$, obtained by executing program segments through the Vizdoom interpreter, along with the candidate program embedding, as inputs.

Table 2: The decoder structure aimed at enhancing program synthesis through environmental contexts.

| Layers | Output |
|---|---|
| 128-2 GATv2Conv LeakyReLU | $L_t \times 256$ |
| 128-2 GATv2Conv | $L_t \times 256$ |
| $3 \times 3$-1-8 Conv BN LeakyReLU | $30 \times 40 \times 8$ |
| $3 \times 3$-1-8 Conv BN LeakyReLU | $30 \times 40 \times 8$ |
| 9600-128 Linear | 128 |
| $3 \times 3$-1-8 Conv BN LeakyReLU | $30 \times 120 \times 8$ |
| $3 \times 3$-1-8 Conv BN LeakyReLU | $30 \times 120 \times 8$ |
| 28800-128 Linear | 128 |
| 256-2 LSTM | 256 |
| 256-$|\mathcal{V}|$ Linear | $|\mathcal{V}|$ |

## A.3  Hyper-parameters

**LGRL** [2]: We employ the identical architecture as the original implementation[1], which utilizes 2D convolution BN ReLU for I/O encoding. We set the kernel size to $d_{k_1} = 3$ and $d_{k_2} = 3$, convolutional stacks to $[64, 64, 64]$, fully-connected stack to $512$, embedding size to $256$, 2 layered LSTM hidden size to $256$, and batch size to $8$. The model is trained using Adam optimizer, with a learning rate of $10^{-4}$, learned syntax penalty of $10^{-5}$.

**SED** [6]: We utilize the I/O encoder architecture, as illustrated in Table 1, based on the original implementation[2]. For the synthesis model, we set the kernel size to $d_{k_1} = 3$ and $d_{k_2} = 3$, convolutional stacks to $[64, 64, 64]$, gradient clip to $5$, warm-up to $40$, bi-directional LSTM hidden size to $256$, and batch size to $8$. The model is trained using SGD optimizer, with a learning rate of $10^{-3}$ and decay rate of $0.5$ after $100000$ steps. For the debugger model, we set the mutate distribution to $[1, 2, 3]$, learning rate to $10^{-4}$, and max beam to $50$, while keeping other parameters the same.

**Inferred Trace** [10]: The architecture and parameters are similar to LGRL, with the addition of an extra $3 \times 3$-1-15 Conv BN LeakyReLU layer and a Linear layer for the encoder to infer execution traces. The decoder also includes a $3 \times 3$-1-8 Conv BN LeakyReLU layer and a Linear layer to incorporate the execution features.

**Latent Execution** [3]: The architecture used is identical to the original one[3]. We have set the embedding size to $1024$, the hidden size of the 2 layered LSTM to $512$, the hidden size of the single-layered MLP to $512$, and the number of attention layers to $2$. Additionally, we have set the gradient clip to $5$, the batch size to $8$, and enabled latent execution. The model has been trained using the SGD optimizer, with a learning rate of $10^{-4}$ and a decay rate of $0.9$ after $6000$ steps.

**Transformer** [12]: In order to facilitate I/O embedding learning, we have utilized the encoder structure (Table 1) on top of the Transformer. The Transformer embedding size has been set to $512$, with 2 attention heads, 2 encoder layers, and 2 decoder layers. The remaining parameters are similar to LGRL.

**EVAPS**[4]: To enhance the quality of candidate programs by incorporating environmental contexts, we have utilized the decoder structure presented in Table 2. We have set the kernel size to $d_{k_1} = 3$ and $d_{k_2} = 3$, the convolutional stacks to $[64, 64, 64]$, the fully-connected stack to $512$, the embedding size to $256$, the hidden size of the 2 layered LSTM to $256$, and the batch size to $4$. Additionally, we have set the batch normalization momentum to $0.1$ and the negative slope of the leakyReLU to $0.01$.

---

[1] https://github.com/bunelr/GandRL_for_NPS

[2] https://github.com/sunblaze-ucb/SED

[3] https://github.com/Jungyhuk/latent-execution

[4] Implementation available at: https://anonymous.4open.science/r/EVAPS-review

The model has been trained using the Adam optimizer, with a learning rate of $10^{-4}$ and a learned syntax penalty of $10^{-5}$.

# B  Additional Experimental Results

## B.1  Dataset Properties

**Overall Synthesis Benchmark.** As delineated in Section 4.1, the dataset is engendered by adhering to the tenets of antecedent studies [2, 4, 6, 11], culminating in $100,000$ unique samples. The mean program sequence length for these instances amounts to $13.37$ tokens, while the average steps necessitated for task completion is $4.59$ steps. The program sequence length spans a range of $5$ to $20$ tokens, and the steps required vary between a minimum of $2$ and a maximum of $13$.

**Task Complexity.** The number of samples in each complexity category is visualized in Figure 1. Overall, the distribution of samples remains equitable, precluding the model from capturing invalid features and generating wrong tokens. The detailed information of each category is presented in Table 3.

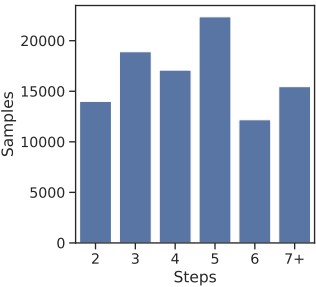

Figure 1: Distribution of the number of samples in each category.

Table 3: Detailed program information of varying levels of complexity.

| Complexity | Program Length | | | Steps | |
|---|---|---|---|---|---|
| | Min | Max | Avg | Avg | Max |
| 2 | 5 | 20 | 11.85 | - | |
| 3 | 6 | 20 | 14.18 | - | |
| 4 | 7 | 20 | 13.67 | - | |
| 5 | 8 | 20 | 13.52 | - | |
| 6 | 9 | 20 | 13.24 | - | |
| 7+ | 10 | 19 | 13.27 | 8.11 | 13 |

## B.2  Additional Results

**Task Complexity.** Table 4 demonstrates the primary outcomes of the EVAPS model in comparison to other techniques when confronted with diverse levels of task complexity. Overall, as the task complexity escalates, EVAPS excels in decomposing tasks into more straightforward actions and exhibiting superior generalization. This underscores the efficacy of utilizing environmental contexts. Meanwhile, SED can still produce comparable outcomes by rectifying errors through execution traces.

Table 4: The average accuracy (with standard deviation) of all methods evaluated on three metrics in six complexity categories, assessed over 5 random seeds. The **best results** are highlighted in bold.

| Complexity | Top-K | Methods | Exact Match | Semantic Match | Generalization |
|---|---|---|---|---|---|
| | Top-1 | LGRL | 10.39% (0.92%) | 67.53% (2.75%) | 66.23% (1.84%) |
| | | SED | 52.72% (1.59%) | 59.22% (1.91%) | 58.83% (1.51%) |
| | | Inferred Trace | 25.97% (0.92%) | 58.44% (1.84%) | 57.14% (2.75%) |
| | | Latent Execution | 9.94% (2.75%) | 68.83% (0.92%) | 68.83% (0.91%) |
| | | Transformer | 35.06% (1.84%) | 44.16% (2.75%) | 42.86% (1.83%) |
| | | **EVAPS** | **68.83%** (7.35%) | **74.03%** (1.32%) | **74.03%** (0.91%) |
| 2 | Top-5 | LGRL | 12.99% (4.59%) | 81.82% (0.92%) | 80.52% (0.92%) |
| | | SED | 66.49% (4.93%) | 73.25% (3.57%) | 72.93% (5.38%) |
| | | Inferred Trace | 33.77% (1.83%) | 72.73% (2.75%) | 70.13% (0.92%) |

| | | | | | |
|---|---|---|---|---|---|
| | | Latent Execution | 15.94% (3.67%) | 72.73% (2.75%) | 71.42% (2.54%) |
| | | Transformer | 58.44% (2.75%) | 70.13% (0.92%) | 67.53% (1.03%) |
| | | **EVAPS** | **83.12%** (6.43%) | **85.71%** (1.84%) | **83.12%** (0.92%) |
| | Top-20 | LGRL | 23.38% (1.83%) | 85.71% (0.92%) | 84.42% (0.92%) |
| | | SED | 69.61% (2.53%) | 81.09% (1.81%) | 81.06% (2.96%) |
| | | Inferred Trace | 44.16% (3.67%) | 81.82% (3.28%) | 79.22% (1.84%) |
| | | Latent Execution | 26.18% (2.75%) | 79.22% (0.92%) | 77.92% (1.08%) |
| | | Transformer | 75.32% (0.91%) | 77.92% (6.43%) | 75.32% (6.42%) |
| | | **EVAPS** | **87.01%** (4.59%) | **88.31%** (1.86%) | **87.01%** (2.75%) |
| 3 | Top-1 | LGRL | 1.92% (1.36%) | 44.23% (6.12%) | 39.42% (5.43%) |
| | | SED | 43.84% (0.75%) | 43.86% (3.01%) | 43.84% (3.96%) |
| | | Inferred Trace | 22.12% (1.36%) | 49.04% (1.39%) | 46.15% (0.68%) |
| | | Latent Execution | 5.91% (1.06%) | **56.73%** (3.40%) | 53.84% (3.39%) |
| | | Transformer | 15.38% (4.76%) | 31.73% (2.04%) | 31.73% (1.36%) |
| | | **EVAPS** | **54.80%** (3.40%) | **56.73%** (6.12%) | **54.80%** (5.44%) |
| | Top-5 | LGRL | 2.88% (2.04%) | 67.31% (1.36%) | 63.46% (0.68%) |
| | | SED | 60.00% (0.75%) | 65.76% (0.94%) | 63.15% (0.79%) |
| | | Inferred Trace | 30.77% (3.40%) | 60.58% (0.68%) | 56.73% (2.72%) |
| | | Latent Execution | 11.01% (0.68%) | 62.50% (5.19%) | 60.58% (5.88%) |
| | | Transformer | 34.62% (3.40%) | 59.62% (5.44%) | 59.62% (4.76%) |
| | | **EVAPS** | **71.15%** (1.36%) | **76.92%** (0.68%) | **75.00%** (0.79%) |
| | Top-20 | LGRL | 7.69% (3.40%) | 76.92% (1.35%) | 74.03% (0.68%) |
| | | SED | 65.38% (0.72%) | 76.92% (3.39%) | 76.73% (4.28%) |
| | | Inferred Trace | 46.15% (0.69%) | 74.04% (0.67%) | 71.15% (1.04%) |
| | | Latent Execution | 24.55% (1.36%) | 73.08% (3.39%) | 71.15% (4.76%) |
| | | Transformer | 59.61% (1.36%) | 81.73% (1.35%) | 80.76% (2.07%) |
| | | **EVAPS** | **80.77%** (2.72%) | **83.65%** (0.68%) | **81.73%** (0.91%) |
| 4 | Top-1 | LGRL | 4.25% (0.75%) | 39.36% (1.50%) | 36.17% (1.50%) |
| | | SED | 34.89% (0.98%) | 34.89% (0.13%) | 34.89% (0.45%) |
| | | Inferred Trace | 19.14% (2.26%) | 39.36% (1.50%) | 36.17% (2.25%) |
| | | Latent Execution | 5.43% (0.75%) | 36.17% (2.26%) | 34.04% (3.01%) |
| | | Transformer | 10.64% (0.75%) | 24.47% (4.51%) | 22.34% (3.44%) |
| | | **EVAPS** | **43.62%** (3.76%) | **46.81%** (2.25%) | **43.62%** (2.56%) |
| | Top-5 | LGRL | 10.64% (3.01%) | 57.45% (0.75%) | 53.19% (0.75%) |
| | | SED | 50.21% (3.85%) | 56.98% (0.57%) | 56.70% (2.07%) |
| | | Inferred Trace | 29.78% (3.01%) | 51.06% (3.08%) | 46.81% (4.51%) |
| | | Latent Execution | 14.05% (0.75%) | 38.30% (1.50%) | 36.17% (2.26%) |
| | | Transformer | 35.11% (5.26%) | 47.87% (4.31%) | 46.81% (5.27%) |
| | | **EVAPS** | **56.38%** (1.50%) | **65.96%** (3.01%) | **62.77%** (2.26%) |
| | Top-20 | LGRL | 15.96% (2.26%) | 67.02% (3.76%) | 64.89% (0.75%) |
| | | SED | 57.87% (3.15%) | 69.16% (3.25%) | 69.00% (4.94%) |
| | | Inferred Trace | 37.23% (3.15%) | 59.57% (1.50%) | 56.38% (2.26%) |
| | | Latent Execution | 21.88% (1.54%) | 54.26% (2.25%) | 52.13% (2.26%) |
| | | Transformer | 43.62% (1.50%) | 62.77% (0.75%) | 60.64% (0.75%) |
| | | **EVAPS** | **68.08%** (0.75%) | **74.47%** (2.25%) | **72.34%** (2.26%) |
| | Top-1 | LGRL | 13.82% (0.57%) | 30.89% (4.02%) | 30.08% (3.45%) |
| | | SED | 40.98% (0.67%) | 40.98% (0.29%) | 40.98% (0.76%) |
| | | Inferred Trace | 23.58% (0.57%) | 34.96% (0.74%) | 34.96% (0.48%) |
| | | Latent Execution | 4.51% (1.72%) | 42.28% (1.15%) | 38.21% (2.30%) |
| | | Transformer | 20.33% (1.15%) | 27.64% (2.30%) | 24.64% (2.29%) |

| | | | | | |
|---|---|---|---|---|---|
| | | | **EVAPS** | **47.97%** (1.15%) | **52.03%** (1.14%) | **51.22%** (1.23%) |

| | | | | | |
|---|---|---|---|---|---|
| | | | **EVAPS** | **47.97%** (1.15%) | **52.03%** (1.14%) | **51.22%** (1.23%) |
| | Top-5 | LGRL | 21.14% (3.45%) | 47.97% (2.87%) | 47.15% (2.30%) |
| | | SED | 57.84% (3.54%) | 58.54% (2.02%) | 58.01% (3.07%) |
| | | Inferred Trace | 31.70% (4.02%) | 50.41% (0.57%) | 49.59% (1.58%) |
| | | Latent Execution | 7.07% (2.87%) | 44.72% (0.57%) | 43.09% (1.72%) |
| | | Transformer | 36.59% (2.53%) | 43.08% (1.72%) | 42.27% (1.76%) |
| | | **EVAPS** | **59.35%** (0.57%) | **66.67%** (1.72%) | **65.85%** (1.65%) |
| | Top-20 | LGRL | 26.83% (2.29%) | 58.54% (5.75%) | 56.91% (5.17%) |
| | | SED | 65.84% (2.84%) | 66.80% (1.94%) | 66.78% (2.53%) |
| | | Inferred Trace | 46.34% (0.75%) | 61.79% (1.15%) | 60.98% (1.14%) |
| | | Latent Execution | 17.82% (3.89%) | 52.85% (2.29%) | 52.03% (2.53%) |
| | | Transformer | 49.59% (1.14%) | 59.35% (1.15%) | 58.54% (1.97%) |
| | | **EVAPS** | **65.85%** (2.87%) | **71.54%** (1.74%) | **69.11%** (1.72%) |
| 6 | Top-1 | LGRL | 11.94% (3.17%) | 29.85% (2.11%) | 25.37% (1.05%) |
| | | SED | 40.60% (0.22%) | 40.60% (1.83%) | 40.60% (2.88%) |
| | | Inferred Trace | 20.90% (1.06%) | 40.30% (2.11%) | 38.81% (3.17%) |
| | | Latent Execution | 3.81% (1.06%) | 32.84% (3.17%) | 31.34% (4.22%) |
| | | Transformer | 8.96% (1.05%) | 22.39% (2.31%) | 19.40% (1.06%) |
| | | **EVAPS** | **49.25%** (2.11%) | **53.73%** (3.16%) | **50.75%** (1.06%) |
| | Top-5 | LGRL | 19.40% (2.11%) | 52.24% (2.11%) | 46.27% (1.07%) |
| | | SED | 57.31% (0.34%) | 62.71% (1.15%) | 61.46% (2.21%) |
| | | Inferred Trace | 38.80% (1.06%) | 64.18% (1.05%) | 59.70% (1.53%) |
| | | Latent Execution | 6.94% (1.05%) | 38.81% (1.06%) | 32.84% (2.11%) |
| | | Transformer | 31.34% (1.06%) | 55.22% (5.28%) | 50.75% (5.26%) |
| | | **EVAPS** | **64.18%** (4.22%) | **76.12%** (2.11%) | **71.64%** (1.05%) |
| | Top-20 | LGRL | 31.34% (3.17%) | 65.67% (1.06%) | 61.19% (1.05%) |
| | | SED | 63.28% (1.44%) | 74.35% (1.70%) | 74.08% (4.86%) |
| | | Inferred Trace | 47.76% (3.17%) | 68.66% (2.11%) | 64.18% (2.17%) |
| | | Latent Execution | 18.16% (1.38%) | 47.76% (2.17%) | 43.28% (2.06%) |
| | | Transformer | 55.22% (1.58%) | 67.16% (3.17%) | 62.69% (3.16%) |
| | | **EVAPS** | **68.66%** (2.07%) | **83.58%** (2.65%) | **79.10%** (1.06%) |
| 7+ | Top-1 | LGRL | 9.41% (0.83%) | 22.35% (0.66%) | 21.18% (0.82%) |
| | | SED | 39.53% (0.83%) | 39.53% (1.91%) | 39.53% (2.34%) |
| | | Inferred Trace | 21.18% (3.32%) | 36.47% (2.49%) | 35.29% (3.27%) |
| | | Latent Execution | 3.18% (0.83%) | 27.06% (2.49%) | 27.06% (2.56%) |
| | | Transformer | 14.12% (3.32%) | 18.82% (0.83%) | 18.82% (0.89%) |
| | | **EVAPS** | **52.94%** (4.99%) | **52.94%** (4.34%) | **49.41%** (3.32%) |
| | Top-5 | LGRL | 21.17% (0.83%) | 54.12% (0.83%) | 51.76% (0.32%) |
| | | SED | 55.53% (0.89%) | 58.85% (1.35%) | 58.85% (1.07%) |
| | | Inferred Trace | 42.35% (2.49%) | 58.82% (1.66%) | 56.47% (0.83%) |
| | | Latent Execution | 7.76% (2.70%) | 31.76% (2.50%) | 31.76% (1.66%) |
| | | Transformer | 32.94% (7.48%) | 42.35% (8.32%) | 42.35% (8.31%) |
| | | **EVAPS** | **65.89%** (0.83%) | **71.76%** (2.50%) | **69.41%** (1.66%) |
| | Top-20 | LGRL | 27.06% (2.49%) | 67.06% (1.56%) | 63.52% (5.82%) |
| | | SED | 68.66% (2.49%) | 71.29% (2.36%) | 71.29% (3.01%) |
| | | Inferred Trace | 56.47% (0.83%) | 70.59% (1.66%) | 67.06% (1.37%) |
| | | Latent Execution | 17.64% (3.32%) | 50.59% (2.31%) | 49.41% (2.46%) |
| | | Transformer | 49.41% (3.25%) | 63.53% (4.99%) | 61.18% (4.16%) |
| | | **EVAPS** | **74.12%** (0.83%) | **77.65%** (1.89%) | **75.29%** (0.81%) |

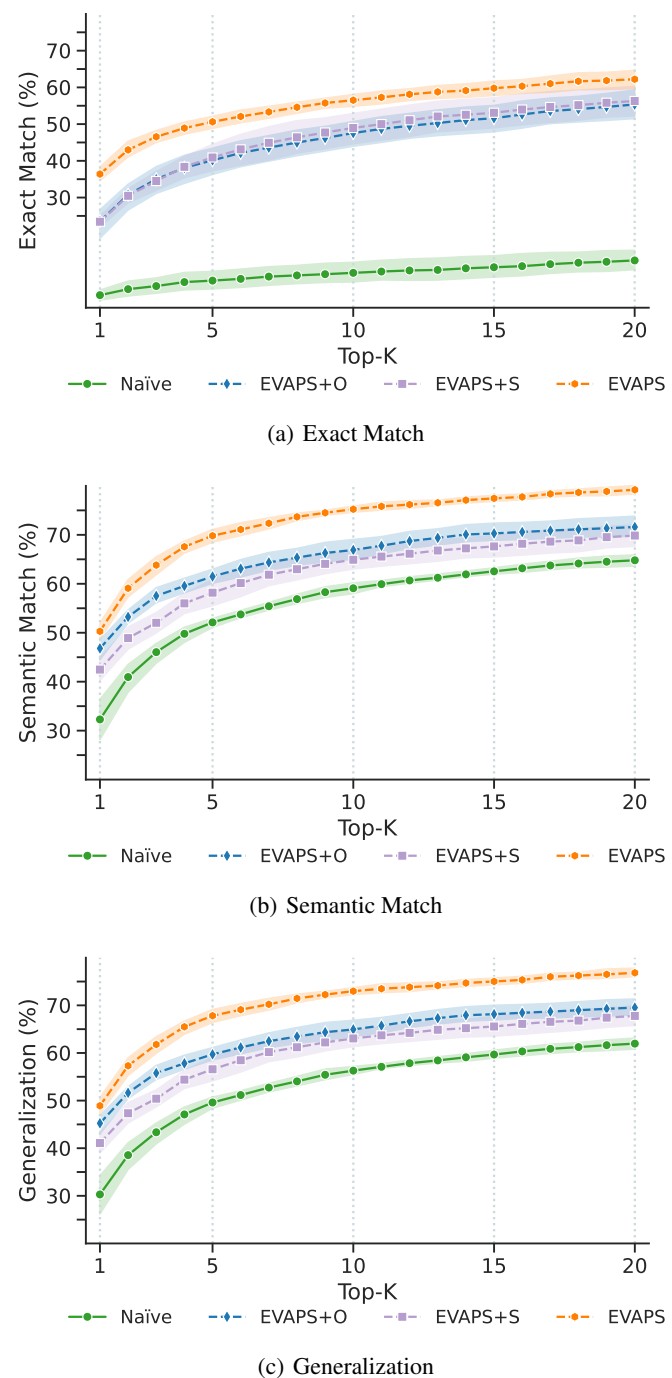

(a) Exact Match

(b) Semantic Match

(c) Generalization

Figure 2: The visualized ablation results, depicting a range from Top-1 to Top-20, and are accompanied by a 95% confidence interval band.

**Ablation Study.** Figure 2 illustrates the comprehensive ablation results evaluated on three metrics. It can be inferred that the generalization ability is enhanced by leveraging partial observations or aligning code symbols, and this improvement is particularly noticeable in predicting exact matched sequences. Furthermore, the interval band for EVAPS is smaller than that of EVAPS+O and EVAPS+S, indicating that the model's stability and robustness are enhanced by incorporating both modules.

## C  Broader Impact

The fundamental concept of utilizing environmental observations and aligning them with code symbols to enhance program synthesis generalization capability can be implemented in actual robotic devices. Although the idea holds promise for real-world scenarios, the current focus is on program generation. We anticipate that the proposed method will not generate any biased or offensive content. However, when gathering observation data from the surroundings, it is imperative to avoid infringing on privacy. Robots are bound to interact with the environment, and to enable the proposed model, environmental data collection is necessary. Typically, the data comprises RGB images that may contain facial data or result in other forms of privacy infringement. Therefore, it is crucial to ensure that the collected environmental data is desensitized before further analysis. We recommend utilizing the proposed algorithm solely for research purposes.