# OpenReview forum: "Enhancing Robot Program Synthesis Through Environmental Context"
_NeurIPS.cc/2023/Conference — NeurIPS 2023 poster_

### Official Review · Reviewer_mJeF · 2023-07-05

**Soundness:** 3 good
**Presentation:** 3 good
**Contribution:** 2 fair
**Rating:** 3
**Confidence:** 4

**Summary:**


This paper introduces the Environmental-context Validated lAtent Program Synthesis framework (EVAPS), which builds upon the SED approach by utilizing a trail-eval-repair loop to enhance program evolution and improve generalization capabilities. EVAPS leverages partial environmental observations by initially obtaining candidate programs from existing synthesizers. By executing these candidate programs and capturing the environmental context before and after each action, EVAPS models both the environmental and syntax contexts concurrently. Through iterative modifications to address semantic conflicts across program tokens, EVAPS aims to correct erroneous program fragments that do not produce the desired output.

The proposed framework was evaluated in the partially observed Vizdoom domain, involving a robot navigating a 3D world and interacting with various elements, including objects and adversaries. The experiment results demonstrate the superiority of the approach over the baselines in accurately modeling program semantic subtleties and effectively resolving potential errors.

**Strengths:**

+ EVAPS synthesizes robot programs operating in partially observed environments (surrounding RGBD image inputs).
+ EVAPS uses a trail-eval-repair loop to iteratively improve program generalizability by rectifying potentially erroneous code segments.
+ EVAPS is capable of handling noisy observations.

A key novelty of EVAPS, when compared to previous program synthesis approaches in the Karel domain, is its ability to handle partial environmental observations. This capability is crucial for synthesizing robot programs, as relying on global environment information is often impractical in real-world robot navigation tasks. EVAPS consists of two essential modules: the partial observation leveraging module, which emphasizes the environmental context of specific program tokens, and the code symbol alignment module, which focuses on modeling the implicit connections between different code segments in semantic spaces. By integrating both syntax and semantic information, these modules facilitate program repair and enhance the effectiveness of EVAPS in generating accurate and meaningful robot programs.

**Weaknesses:**

The title of the paper may appear to overstate the actual contribution made by the research.  The choice of the Vizdoom environment as a benchmark for evaluating Robotic Program Synthesis in this paper raises some concerns. The evaluation of EVAPS in the navigation-focused Vizdoom environment does appear to have limitations. While efficient navigation is undoubtedly an important robotic task, it may not fully represent the diverse range of tasks in robotics, particularly those involving object manipulation in fully observable environments. It would be valuable to explore the applicability of EVAPS in synthesizing robot programs for other more realistic environments like Miniworld and AI2-Thor, which involve long-horizon vision-based tasks. Assessing the performance of EVAPS in such contexts would provide a more comprehensive evaluation of its capabilities. Given the current state of the paper, it remains uncertain whether EVAPS is adequately equipped to handle the challenges posed by complex robotic navigation tasks. To determine the true extent of EVAPS' applicability across a broader range of robotic tasks and environments, additional research and experimentation are necessary.

Nevertheless, I do believe this paper contributes to the program synthesis community. The comparative evaluation of EVAPS against the baselines, when provided with partial observations, showcases its superior performance, thereby highlighting the significance of the observation leveraging module and the code symbol alignment module. However, there remains a lingering uncertainty regarding the practical implications of this demonstrated value in the context of robot control.

**Questions:**

EVAPS seems incremental to SED. To gain a comprehensive understanding of EVAPS' contribution, it would be beneficial to apply the framework to the Karel domain, where the baselines were evaluated. This approach would effectively demonstrate the effectiveness of the observation leveraging module and the code symbol alignment module. Is there any evidence or indication from the authors regarding the performance of EVAPS specifically in the Karel domain?

**Limitations:**

The paper presents an approach for robot program synthesis. However, the evaluation falls short in providing a convincing argument on how EVAPS can be effectively applied to learn robot-control policies, as the experiments are confined to a relatively simple video game environment. Further investigation and experimentation in more complex and realistic robotic scenarios would be necessary to demonstrate the practical utility and effectiveness of EVAPS in learning robot-control policies.

---

> ### Author Rebuttal · Authors · 2023-08-10
>
> Thanks for the insightful comments and constructive suggestions. We summarise the issues pointed out and address them in the following:
>
>
> > A gap between the title and experiment environment in which this work is evaluated
>
> Thanks for pointing this out. In this work,  we explore the topic "enhancing robotic program synthesis with the execution environment" and believe that it will complement traditional program synthesis and is worthwhile exploring. Considering navigation is one of the most common and important tasks in robotics, we evaluate our work on navigation tasks in robotics. To be accurate, we will make our paper title specific to navigation tasks in the revision.
>
> > Choice of experiment environments
>
> As the reviewer pointed out, there exist several commonly used execution environments for evaluating approaches in robotics including Karel, Vizdoom, Miniworld and AI2-Thor. We went through the following process to choose our experiment environment.
>
> We conducted a review of robotic program synthesis works published in the recent 5 years (the most related works are listed below [1-9]) and found that they use either Karel or Vizdoom as the experiment environment.
>
> In the end, we chose Vizdoom over Karel as our experiment environment since tasks in Vizdoom involve more actions and scenarios are more complex. The differences between them are shown as follows. Karel is a 2D grid world involving 5 marker actions primitives and 5 perceptions primitives for obstacle detection. Vizdoom is a 3D semi-realistic simulation environment involving 7 interactive action primitives and 6 perception primitives [3].
>
> |  Environment   | Dimension  | DSL space (action,perception) | State Representation | Task Type
> |  ----  | ----  | ----  | ----  | ----  |
> | Vizdoom | 3D | ( 7 , 6 ) | 120 x 160 x 3 | Navigation |
> | Karel  | 2D | ( 5 , 5 ) | 8 x 8 x 16 | Navigation |
>
> In general, Vizdoom covers Karel in terms of environmental dimensions, the complexity of state representation as well as action and perception space. Thus, we chose Vizdoom as our execution environment.
>
> > Indication of the performance of our approach in the Karel domain
>
> As discussed above, Vizdoom is sort of an upgraded version of Karel. Relevant actions and perceptions in Karel are contained in Vizdoom. Most of the scenarios and tasks in Karel can be found in Vizdoom. Since our approach outperforms the baselines in Vizdoom, it is reasonable for us to believe consistent results can be achieved in the Karel domain.
>
> Reference
>
> [1] Duan, Xuguang, et al. "Watch, reason and code: Learning to represent videos using program." In Proceedings of  ACM MM, 2019.
>
> [2] Dang-Nhu, Raphaël. "PLANS: Neuro-symbolic program learning from videos." In Proceedings of NeurIPS, 2020.
>
> [3] Sun, Shao-Hua, et al. "Neural program synthesis from diverse demonstration videos." In Proceedings of ICML, 2018.
>
> [4] Gupta K, Christensen P E, Chen X, et al. "Synthesize, execute and debug: Learning to repair for neural program synthesis". In Proceedings of NeurIPS, 2020.
>
> [5] Chen, Xinyun, Dawn Song, and Yuandong Tian. "Latent execution for neural program synthesis beyond domain-specific languages." In Proceedings of NeurIPS, 2021.
>
> [6] Manchin, Anthony, et al. "Program Generation from Diverse Video Demonstrations." arXiv preprint arXiv:2302.00178 (2023).
>
> [7] Chen, Xinyun, Chang Liu, and Dawn Song. "Execution-guided neural program synthesis." In Proceedings of ICLR, 2018.
>
> [8] Shin, Eui Chul, Illia Polosukhin, and Dawn Song. "Improving neural program synthesis with inferred execution traces." In Proceedings of NeurIPS, 2018.
>
> [9] Trivedi, Dweep, et al. "Learning to synthesize programs as interpretable and generalizable policies." In Proceedings of NeurIPS, 2021.

---

> > ### Comment · Reviewer_mJeF · 2023-08-14
> > **Thanks for your response**
> >
> > I appreciate the authors' response. However, I am still not convinced that the video game experiments sufficiently demonstrate the suitability of EVAPS for synthesizing robot programs in visually realistic navigation environments such as Miniworld and AI2-Thor, which entail navigating complex and partially observable long-horizon robot tasks. It would be advisable for the authors to assess the applicability of EVAPS across a wider spectrum of navigation tasks and environments. With that being mentioned, even if the title is narrowed down to navigation tasks, I would still argue that the evaluation does not provide support for it.
> >
> > Regarding program synthesis for robot navigation with partial observability, it looks like at least the benchmarks from the following papers are more challenging than the set of tasks explored in this paper.
> >
> > [1] Cao, Yushi, et al. "GALOIS: Boosting Deep Reinforcement Learning via Generalizable Logic Synthesis." Advances in neural information processing systems (2022).
> >
> > I would keep my score based on the above discussion.

---

> > > ### Author Response · Authors · 2023-08-15
> > > **Response to Reviewer mJeF's Comment**
> > >
> > > Thank you very much for your further comments. As we understand, the remaining concern is the choice of experimental environment.
> > >
> > > We agree with the reviewer that experiments should be conducted in a more realistic environment. This is the reason we chose VizDoom as our experimental environment.
> > >
> > > 1. VizDoom is a more realistic environment. In fact, VizDoom contains more challenging scenarios than the suggested environments:
> > >
> > >     - VizDoom is more complex compared to MiniGrid [1] used in the GALOIS [2] work pointed out by the reviewer:
> > >
> > >         |  Environment   | Dimension  | Action Primitives | State Representation
> > >         |  ----          | ----       | ----              | ----
> > >         | VizDoom       | 3D         | 7                 | 120 x 160 x 3
> > >         | MiniGrid        | 2D         | 5\*         | 20 x 20 (Used in GALOIS [2])
> > >
> > >         *Note\*: Only a maximum of 5 actions primitives are simultaneously available in a single scenario.*
> > >
> > >         Considering all four scenarios (DoorKey, BoxKey, UnlockPickup, Multiroom) evaluated in the GALOIS [2] work, "equivalent" scenarios can be found in VizDoom.
> > >
> > >     - VizDoom is more complex than MiniWorld [3] suggested by the reviewer. As described in the README.md of the official repo [4], MiniWorld can be seen as a simpler alternative to VizDoom or DMLab. For your convenience, we brought relevant text here:
> > >         >  MiniWorld is a minimalistic 3D interior environment simulator for reinforcement learning & robotics research. It can be used to simulate environments with rooms, doors, hallways and various objects (eg: office and home environments, mazes). *MiniWorld can be seen as a simpler alternative to VizDoom or DMLab.* It is written 100% in Python and designed to be easily modified or extended by students.
> > >
> > > 2. VizDoom is the commonly adopted experimental environment by the recent robotic program synthesis works [5, 6, 7, 8] according to our literature review reported in the previous response.
> > >
> > > Hopefully, this response can address your concern.
> > >
> > > References
> > >
> > > [1] Maxime Chevalier-Boisvert, Lucas Willems, and Suman Pal. "Minimalistic gridworld environment for
> > > openai gym.", 2018.
> > >
> > > [2] Cao, Yushi, et al. "GALOIS: boosting deep reinforcement learning via generalizable logic synthesis." In Proceedings of NeurIPS, 2022.
> > >
> > > [3] Chevalier-Boisvert, Maxime, et al. "Minigrid & Miniworld: Modular & Customizable Reinforcement Learning Environments for Goal-Oriented Tasks." CoRR, 2023.
> > >
> > > [4] Chevalier-Boisvert, Maxime, "MiniWorld: Minimalistic 3D Environment for RL & Robotics Research", GitHub repository, 2018.
> > >
> > > [5] Manchin, Anthony, et al. "Program Generation from Diverse Video Demonstrations." arXiv preprint arXiv:2302.00178, 2023.
> > >
> > > [6] Dang-Nhu, Raphaël. "PLANS: Neuro-symbolic program learning from videos." In Proceedings of NeurIPS, 2020.
> > >
> > > [7] Duan, Xuguang, et al. "Watch, reason and code: Learning to represent videos using program." In Proceedings of ACM MM, 2019.
> > >
> > > [8] Sun, Shao-Hua, et al. "Neural program synthesis from diverse demonstration videos." In Proceedings of ICML, 2018.

---

> > > > ### Comment · Reviewer_mJeF · 2023-08-17
> > > > **Clarification**
> > > >
> > > > > MiniWorld can be seen as a simpler alternative to VizDoom or DMLab. It is written 100% in Python and designed to be easily modified or extended by students.
> > > >
> > > > I'd like to emphasize that MiniWorld is considered a simpler alternative only because of its lightweight computation demand, not because its tasks are simpler. In fact, when comparing the complexity of tasks in MiniWorld (including Galois examples) with your benchmarks, they are notably *way more* challenging due to their **long-horizon** nature. These tasks entail hundreds or even thousands of control steps. In your context, the maximum number of control steps for task completion is 13, which appears to be quite limited for realistic robot tasks.
> > > >
> > > > Cited from my original review
> > > > > It would be valuable to explore the applicability of EVAPS in synthesizing robot programs for other more realistic environments like Miniworld and AI2-Thor, which involve long-horizon vision-based tasks.
> > > >
> > > > The author response does not address my concern about whether EVAPS generalizes to long-horizon robotic navigation tasks. Because of this, I would recommend rejection. However, if the other reviewers are enthusiastic about it, I will not fight for not accepting it.

---

> > > > > ### Author Response · Authors · 2023-08-21
> > > > > **Thanks**
> > > > >
> > > > > Thanks for the clarification.
> > > > > We will incorporate your suggestions in the revision.

---

### Official Review · Reviewer_wkgC · 2023-07-05

**Soundness:** 2 fair
**Presentation:** 1 poor
**Contribution:** 2 fair
**Rating:** 5
**Confidence:** 3

**Summary:**

The paper claims that global observation in robot program synthesis is not achievable, so it proposes to use partial observation. And it learns an observation embedding module and a semantic-grammatical alignment module to repair the candidate programs that can increase the accuracy and generalization of robot program synthesis.

**Strengths:**

1. The problem this paper focused on is important to the application of PS as the perfect and complete observable environment is often not available in the real world.

2. The experiment setting and analysis are clear and comprehensive. The paper analyses the effect of multiple aspects factors in the in-perfect observation of the proposed method.

**Weaknesses:**

[W1]. The key weakness is the choice of baseline methods. As a program repair method, it’s better to compare with methods that have a fair setting rather than program generation methods. There are other program repair methods that utilize trajectory to enhance performance. Moreover, the baseline methods are too old (most before 2020).

[1] Execution-guided neural program synthesis

[2] Write, execute, assess: Program synthesis with a REPL

[2] Improving neural program synthesis with inferred execution traces


[W2]. Needs more explanation about the difference between global environment information and partially observable environments.

[W2.1] As using pre-observation and post-observation which are related to specific program segments to enhance the information contained in the embedding is a general trick. The PS methods which take global environment information as input can also learn such representation to enhance accuracy and generalization. What is the key connection between the proposed method and partially observable environments?

[W2.2] Needs some insightful evaluation in the results section to demonstrate the relation between the main motivation (partial observable environment) and the proposed architecture.

[W3]. There are some typos, like line 203, citation 41 is not the baseline

**Questions:**

Please see weaknesses.

---

> ### Author Rebuttal · Authors · 2023-08-10
>
> Thanks for the insightful comments and constructive suggestions. We summarise the issues pointed out and address them in the following:
>
> ---
>
> > Choice of baseline
>
> We conducted a review of robotic program synthesis works published in the recent 5 years and identified SED proposed in 2020 as the state-of-the-art technique in robotic program synthesis. In the evaluation, we compared our approach with SED as shown in Section 4.2. The experiment results show our approach significantly outperforms SED.
>
> Regarding traditional program repair approaches, we are indeed aware of a branch of rich works in the program repair area, which can be classified into three categories:
> - Search-based. This type of approach considers program repair as a search problem, exploring the space of all possible program candidates to identify one that satisfies the given weak specification, i.e., test cases. The representative work of this area is GenProg [1].
> - Semantic-based. This type of approach extracts semantic information from the program under repair (typically represented as path constraints) and then generates patches by solving those constraints. The representative work is SemFix [2].
> - Learning-based. This type of approach leverages a number of patches generated by developers to learn a model that repairs programs. An earlier and representative work is Prophet [3].
>
> Those approaches depend on high-quality test suites to validate patch candidates, which are unavailable in the setting our approach is targeted at, i.e., Vizdoom. Besides, Karel and Vizdoom are designed for specific purposes. Complex features in programs that traditional program repair approaches rely on can be abstracted away. These pose challenges for applying traditional program repair approaches to such domains.
> For these reasons, robotic program synthesis work with a focus on program repair SED [4] also does not have a comparison with traditional program repair approaches.
>
> > More explanation on the difference between global execution environment and partial execution environment
>
> In the context of robotic systems, "partial environment" pertains to the immediate surroundings that the robot is capable of perceiving through its sensory devices, whereas the concept of "global environment" encompasses a comprehensive understanding of the entire scenario, including elements beyond the robot's perceptual capabilities [5, 6]. In the real world, global observations are typically unavailable and only partial observations are available. Thus, our approach is focused on partial observations for enhancing program synthesis.
>
> > The key connection between the proposed approach and partially observable environments
>
> The key connection between the proposed approach and observable environments is our unique design that establishes a connection between the program execution context and the partially observable environment. As shown in lines 127-133, our approach takes the execution context of statement $S$ and the partially observable environment of statement $S$ as a unit for model training. This enables the combination of program syntax and the corresponding partially observable environment to predict a token, thereby enhancing the accuracy of program synthesis.
>
> > Evaluation of the key connection between the proposed approach and partially observable environment
>
> We conducted a study to evaluate the effectiveness of our design in the experiment. As shown in Table 2, EVAPS achieves the best performance and its performance significantly goes down when the unique design is taken away. “EVAPS+O” and “EVAPS+S” are variants of EVAPS without connecting the program execution context and the partially observable environment and they both underperform EVAPS.
>
> > Typos
>
> Thank you for your feedback, we will carefully proofread the manuscript and correct errors pointed out.
>
> References
>
> [1] Le Goues, Claire, et al. "Genprog: A generic method for automatic software repair." Ieee transactions on software engineering, 2011.
>
> [2] Nguyen, Hoang Duong Thien, et al. "Semfix: Program repair via semantic analysis." In Proceedings of ICSE, 2013.
>
> [3] Long, Fan, and Martin Rinard. "Automatic patch generation by learning correct code." In Proceedings of POPL, 2016.
>
> [4] Gupta K, Christensen P E, Chen X, et al. "Synthesize, execute and debug: Learning to repair for neural program synthesis". In Proceedings of NeurIPS, 2020.
>
> [5] Chen, Ci, et al. "Motion planning for heterogeneous unmanned systems under partial observation from uav." In Proceedings of IROS, 2020.
>
> [6] Katsumata, Yuki, et al. "Map completion from partial observation using the global structure of multiple environmental maps." Advanced Robotics, 2022.

---

> > ### Comment · Reviewer_wkgC · 2023-08-19
> > **Thanks for your rebuttal**
> >
> > 1. Thanks for the authors' rebuttal, the concern about baselines is addressed. I suggest the author add these discussions to the paper.
> > 2. However, I still think that the core design of the proposed method and the problem of the partially observed env lack some key and reasonable connection.  I will keep my score. Thanks for the rebuttal.

---

> > > ### Author Response · Authors · 2023-08-21
> > > **Thanks**
> > >
> > > Thanks for your comments.
> > > We will add the discussion about the baselines and elaborate more on the key connection between the proposed method and the partially observed environment in the revision.

---

### Official Review · Reviewer_2nvQ · 2023-07-05

**Soundness:** 4 excellent
**Presentation:** 4 excellent
**Contribution:** 3 good
**Rating:** 6
**Confidence:** 2

**Summary:**

This paper proposes EVAPS to enhance robotic program synthesis by integrating parietal environmental observations. Specifically, EVAPS utilizes both the environmental context leveraging module and the code symbol alignment module to improve its ability to rectify semantically erroneous program segments and generalize across various tasks. Comprehensive experiments on the partially observed Vizdoom benchmark demonstrate its superior performance over other baselines across various tasks. Overall, this is a well-written paper and its extensive experiments and ablation studies verify the effectiveness of the proposed method. I would be leaning to accept this paper.

**Strengths:**

* The proposed method is technically sound and well motivated.
* This paper is well written and well structured. It provides clear formulations and an overview figure to well explain the method.
* The experimental evaluation is rather comprehensive and provides convincing results to demonstrate the effectiveness of the proposed method.


**Weaknesses:**

 I did not identify any weaknesses for this paper as I am not familiar with the task of robotic program synthesis.

**Questions:**

Can you shed some light on the limitations of the proposed method?

**Limitations:**

This paper just discussed the limitations very lightly. I would like to see more discussion on its limitations.

---

> ### Author Rebuttal · Authors · 2023-08-10
>
> Thanks for acknowledging our contributions. We address your comments in the following.
>
> > Limitation
>
> Since our approach is based on self-supervised training, it relies on the quality of the data used for training.
> However, data quality is a common issue in this field.
>
> Moreover, we anticipate some practical difficulties during the transition from the simulation environment to the real world. It is labor-intensive to collect enough training environment data from the real world.

---

> ### Comment · Reviewer_2nvQ · 2023-08-18
> **Official comment by Reviewer 2nvQ**
>
> Thanks for the response. I will stick to my leaning to accept recommendation.

---

> > ### Author Response · Authors · 2023-08-21
> > **Thanks**
> >
> > Thank you for taking the time to respond.

---

### Official Review · Reviewer_emor · 2023-07-05

**Soundness:** 3 good
**Presentation:** 3 good
**Contribution:** 3 good
**Rating:** 6
**Confidence:** 3

**Summary:**

The paper proposes Environmental-context Validated lAtent Program Synthesis framework (EVAPS). a program synthesis model that generates executable program for robotic programming, evaluated in the vizdoom environment. It initially obtains candidate programs using other available synthesizers, then performs program repair by executing the candidate program and collect resulted partial environmental observations. It outperfoms a range of prior works for viszoom program synthesis, and demonstrated robustness again observation noises and task complexity.

**Strengths:**

- using the aid of partially observed environments for program synthesis and repair is novel and reasonable
- the proposed framework is sound, and the design choices for utilizing observations, and using a graph structure to aggregate environmental and syntactic information flow is convincing
- extensive experiments

**Weaknesses:**

- the writing of the paper can be improved, by reducing repetitive and inconsistent adjectives
- do the baselines use program repair? Does any of them rely on environmental observations?
- the assumption of executing candidate programs makes more sense when a privileged simulation environment is available, and when real-time control is not needed. Can the author provide further justification regarding this? How would you expect to transfer to real world?
- the author claim that EVAPS shows is more robust agains noise in the conclusion, but there's no comparison with baseline  on this matter in the experiment section.

**Questions:**

see weeknesses.

**Limitations:**

The major limitation i see in this framework is the assumption of aqcuiring partial observations and executing program candidates in the environment. These assumption is not valid in real-world real-time control. More justifications on this would be very helpful.

---

> ### Author Rebuttal · Authors · 2023-08-10
>
> Thanks for your constructive comments. We address your comments in the following.
>
> > Writing of the paper can be improved
>
> Thanks for your comments. We will proofread the manuscript carefully.
>
> > Do the baselines use program repair? Does any of them rely on environmental observations?
>
> In baselines, SED [1] uses execution feedback to fix the program. It first produces initial programs using the neural program synthesizer component, then utilizes a neural program debugger to iteratively repair the generated programs. SED also uses environmental observation for training the neural program debugger.
>
> However, there exist significant distinctions between SED and our approach:
> - SED's execution feedback relies on a global perspective, which is only available in some special cases. In contrast, our approach embraces partial observation, which is more achievable in the real world.
>
> - SED treats the program as a whole for training while our approach pays more attention to the execution context.
>
> - Our approach establishes a connection between the program execution context and the partial observable environment and this connection significantly enhances the performance of our approach.
>
> > How would you expect to transfer to the real world?
>
> The partial observation that our approach relies on is often available in practice, so it is theoretically feasible to apply our approach to the real world.
> Certainly, we anticipate some practical difficulties during the transition from the simulation environment to the real world. It is labor-intensive to collect enough training environment data from the real world.
>
>
> > The author claim that EVAPS shows is more robust against noise in the conclusion, but there's no comparison with the baseline on this matter in the experiment section.
>
> To clarify, we are not evaluating if EVAPS achieves better robustness over other baselines. Instead, we aim to evaluate how EVAPS performs in an environment with noise. The result shows that our approach demonstrates relatively good robustness.
>
>
> Reference
>
> [1] Gupta K, Christensen P E, Chen X, et al. "Synthesize, execute and debug: Learning to repair for neural program synthesis". In Proceedings of NeurIPS, 2020.

---

> > ### Comment · Reviewer_emor · 2023-08-19
> >
> > Thanks for your rebuttal. I will keep my score.

---

> > > ### Author Response · Authors · 2023-08-21
> > > **Thanks**
> > >
> > > Thank you for taking the time to respond.

---

### Official Review · Reviewer_ivtK · 2023-07-06

**Soundness:** 4 excellent
**Presentation:** 4 excellent
**Contribution:** 3 good
**Rating:** 7
**Confidence:** 3

**Summary:**

This paper proposes an approach for program synthesis in robotic domains where the environment context can provide valuable insight for what the correct program should be. The approach takes candidate programs, executes them to get a trace of observations, and then passes the program and trace through a combination of local and global feature extraction with neural networks. The output is a refined program. The approach is trained and tested on randomly generated Vizdoom programs and is substantially better than the alternatives. They also have an ablation study for the contribution of the two types of feature extraction and a study of the approach's noise tolerance.

**Strengths:**

The results are good. The description of the approach is very clear and easy to understand. Some details are missing but overall it's clear. Vizdoom seems like a popular domain and their approach performs much better than the tested alternatives.

The introduction does a great job of describing the intuition behind incorporating observation feedback into the program proposal.

I don't have a lot to say about the paper. It is mostly a standard type of paper of applying a new, well-thought out approach to an existing domain. The approach seems sound and the results are good. The main insight is incorporating the environmental feedback to improve the program, which is similar to execution-guided synthesis.

**Weaknesses:**

The paper doesn't have any glaring weaknesses. As usual, it could be strengthened by applying the approach to another benchmark.

My main concern is that the gist of the idea seems very similar to work on execution-guided synthesis. But, the approach is shown to be much better than SED on the vizdoom benchmark, so there must be something more here.

# Addressing rebuttal
I have read the authors rebuttal. In particular, they satisfactorily address my main concern about similarity to other work, which I have no further concerns about.

**Questions:**

Can SOTA approaches on Karel be applied to Vizdoom? If so, how do they compare? It seems like the intuitive reason this approach works is similar to the idea behind execution-guided synthesis work applied to Karel, so that could be a good comparison to include.

Where do the program candidates come from before refinement? I don't recall the paper explaining this. This is a big confusion of mine.

I would prefer to have the description of the Vizdoom domain in the evaluation section rather than the preliminary. A figure showing an example task would be helpful too.

---

> ### Author Rebuttal · Authors · 2023-08-10
>
> Thank you for your meticulous review of our paper and for acknowledging our contribution.
> Please see below our responses to your comments.
>
> ---
>
> > Can SOTA approaches on Karel be applied to Vizdoom?
>
> > My main concern is that the gist of the idea seems very similar to work on execution-guided synthesis.
>
> In general, approaches on the Karel domain can be applied to the Vizdoom domain.
> We conducted a review of robotic program synthesis works published in the recent 5 years and identified SED [1] proposed in 2020 as the SOTA approach, which we have evaluated in the paper.
>
> However, there exist significant distinctions between SED and our approach:
>
> - SED's execution feedback relies on a global perspective, which is only available in some special cases. In contrast, our approach embraces partial observation, which is more achievable in the real world.
>
> - SED treats the program as a whole for training while our approach pays more attention to the execution context.
>
> - Our approach establishes a connection between the program execution context and the partial observable environment and this connection significantly enhances the performance of our approach.
>
>
> > Where do the program candidates come from before refinement?
>
> In lines 63-64, we have mentioned that "*EVAPS initially obtains candidate programs through existing synthesizers*".
> Specifically, the candidate programs used in the paper are generated by the same approach used in SED.
> We will elaborate more in the revision.
>
> > I would prefer to have the description of the Vizdoom domain in the evaluation section rather than the preliminary. A figure showing an example task would be helpful too.
>
> Thank you for your suggestion, we will revise the manuscript following your suggestion.
>
>
> References
>
> [1] Gupta K, Christensen P E, Chen X, et al. "Synthesize, execute and debug: Learning to repair for neural program synthesis". In Proceedings of NeurIPS, 2020.

---

> > ### Comment · Reviewer_ivtK · 2023-08-14
> > **Response**
> >
> > Thank you for your response and addressing my comments and concerns.

---

> > > ### Author Response · Authors · 2023-08-21
> > > **Thanks**
> > >
> > > Thank you for taking the time to respond.

---

### Decision · Program_Chairs · 2023-09-21

**Decision:**

Accept (poster)

**Comment:**

The paper proposes an approach for program synthesis in robotic domains that extends the traditional programming by example paradigm with an automated repair step that observes the behavior of the synthesized program and proposes a corrected program based on that.
The idea is original and seems to perform better than a number of state-of-the-art baselines. In particular, the paper builds on prior work on the Synthesize-execute-debug approach from prior program synthesis work, but the experiments show that it performs significantly better.

The main concern from one of the reviewers was that the paper is oversold in its potential as a robotics contribution, since the tasks are too short for what is considered state-of-the art in robotics. Nevertheless, I think this is a valuable contribution for program synthesis, and a clear improvement over the state-of-the-art.